# HER2 Amplification in p53-Mutated Endometrial Carcinomas

**DOI:** 10.3390/cancers15051435

**Published:** 2023-02-24

**Authors:** Ambre Balestra, Denis Larsimont, Jean-Christophe Noël

**Affiliations:** 1Department of Gynecology, ULB-Erasme, HUB, 808 Route de Lennik, 1070 Bruxelles, Belgium; 2Department of Pathology, Institut Bordet, ULB-Erasme, HUB, 808 Route de Lennik, 1070 Bruxelles, Belgium; 3CUREPATH, Rue de Borfilet, 12A, 6040 Jumet, Belgium

**Keywords:** endometrial carcinoma, p53, mutation, HER2, amplification, targeted therapies

## Abstract

**Simple Summary:**

Endometrial cancers with p53 mutations tend to recur and develop metastases more frequently. From this point of view, the detection of new potential therapeutic targets such as HER2 is particularly interesting. In this study, we demonstrated that overexpression of HER2 (++ or +++) was present in 31.4% of p53-mutated endometrial carcinomas and that among these, 36.3% showed amplification of the HER2 gene by CISH (Chromogenic In Situ Hybridization). These results suggest that various anti-HER2 agents could constitute interesting future treatments in this type of cancer.

**Abstract:**

p53-mutated endometrial carcinomas tend to recur and develop distant metastases. Therefore, the detection of new potential therapeutic targets such as HER2 is particularly interesting. In this retrospective study, which considered over 118 endometrial carcinomas, the p53 mutation was detected in 29.6% of cases. In these cases, the HER2 protein profile was studied via immunohistochemistry, and an overexpression of HER2 protein (++ or +++) was noted in 31.4%. The CISH technique was used in these cases to determine if gene amplification was present. In 18% of cases, the technique was not conclusive. Amplification of the HER2 gene was observed in 36.3% of cases and 36.3% of cases showed a polysomal-like aneusomy for centromere 17. Amplification was found in serous carcinomas, clear cell carcinomas and carcinosarcomas, highlighting the future potentiality of HER2-targeted therapies in these variants of aggressive carcinomas.

## 1. Introduction

More than 400,000 women worldwide were affected by endometrial cancer in 2020. The mortality rate of endometrial cancer is estimated to increase by 1.9% per year. The morbidity rate is higher in North America and Western Europe, which could be explained by a lifestyle marked by risk factors for endometrial cancer such as obesity [1].

In 2020, endometrial cancer was the fourth most common tumor in Europe with an incidence of 12.9 to 20.2 per 100,000 women. It is the most frequent gynecologic cancer in Europe [1]. Approximately 1500 cases are diagnosed every year in Belgium, and its incidence of 10.8/100.000 is increasing, in part due to the growing prevalence of obesity [2].

It is established that a high body mass index (BMI) is a modifiable risk factor for endometrial cancer, and it is a more important risk factor for endometrial cancer than for any other cancer [2].

Obesity is a risk factor for endometrial cancer because adipocytes convert androgens into estrogens and estrogens stimulate endometrial proliferation and therefore potentially the development of hyperplasia and cancer. In addition, hyperglycemia and insulin resistance, which are associated with obesity, promote IGF-1 signaling abnormalities, leading to increased cell proliferation [1].

Initially, endometrial cancers were classified into two categories: type I, which represents approximately 80% of cases with a hormone-dependent profile, and type II for the remaining 20% of cases presenting a hormone-independent profile and a poor prognosis [3].

The Cancer Genome Atlas of molecular analyses for different histologic types of endometrial cancers reveled four tumoral subtypes, each associated with a different survival profile and of great interest for prognosis, namely: POLE (Polymerase DNA Epsilon Catalytic Subunit)-mutated tumors; MMR (Mismatch Repair)-stable tumors; MMR-instable tumors; and p53-mutated tumors [4].

This new classification, according to the ESGO (European Society of Gynaecological Oncology), ESTRO (European Society for Radiotherapy and Oncology) and ESP (European Society of Pathology) 2020 recommendations, allows some modulations of the therapeutic schemes, adding to its therapeutic prognosis value [5]. Therefore, some patients could benefit from less aggressive therapies than those suggested under the former classification.

P53-mutated tumors are classically associated with more aggressive tumor histological types and a pejorative prognosis, such as high-grade endometrioid carcinomas, serous carcinomas and carcinosarcomas [4].

The p53 protein is normally involved in the preservation of the cellular genome by blocking the replication of cells containing genomic alterations. It is a beneficial element at the cellular level; therefore, one can understand how important it is for this element to stay active. If a mutation occurs, efficiency decreases or is totally annihilated according to the mutation type. This protein is found to be mutated in 2/3 of cancers, preventing apoptosis when genetic abnormalities are present and thus encouraging cancer survival. On the endometrial cancers’ side, this mutated protein is found in 20 to 40% of cases. One of the characteristics of the mutated p53 protein is that its half-life is increased, and therefore it can be detected by immunohistochemistry [6].

The p53 protein “wild type” has a short half-life and is rapidly degraded after activation; thus, it is kept at low levels in the cell. This mechanism is mainly due to MDM2 and MDM4, which decrease the transcriptional activity of the p53 protein, in particular by making it the target of degradation by proteasomes. When the p53 protein is mutated, this cascade is broken either because the mutated p53 protein does not have the ability to activate MDM2 or because MDM2 becomes a stimulator of p53 protein transcription.

In both cases, the p53 protein accumulates in the cell and thus becomes detectable via immunohistochemistry [7].

The immunohistochemical profile of the p53 protein in endometrial carcinomas has been classified into four patterns [8]:(a)“Nul type”: complete absence of immunohistochemical detection of the protein in tumor cells (with positive internal control).(b)“Wild type”: nuclear immunohistochemical reactivity of low and/or moderate and/or diffuse intensity detected in less than 80% of tumor cells.(c)“Mutant”: strong and diffuse immunohistochemical reactivity detected in 80 to 100% of cells.(d)“Cytoplasmic”: significant cytoplasmic reactivity that is associated with variable nuclear reactivity but does not reach the high and diffuse detection threshold.

The histochemistry of the p53-mutated protein is of interest, but we will refrain from going too much in depth on the subject since it is not the aim of this study.

As p53 mutations in tumors tend to recur more frequently and develop distant metastases within five years in 50% of cases, it is of great interest to determine the presence of potential targets for future targeted therapies, which be of some benefit in the treatment of these varieties of endometrial cancer, whether or not they are targeted in conventional radio/chemotherapy [9].

In particular, it has recently been suggested that overexpression of the HER2 protein may be observed in approximately 25–30% of endometrial serous carcinomas, which are typically p53-mutated and therefore could constitute a potential therapeutic target (by analogy with the treatment regimen already applied in breast cancer) [10,11].

HER2 is a transmembrane receptor involved in cell proliferation. When the HER2 receptor is activated, it stimulates signaling pathways that lead to oncogenic transformation through cell survival, proliferation, angiogenesis and metastasis. HER2 has no known ligand and can therefore be constitutionally activated. In addition, when HER2 is overexpressed, it forms heterodimers with other members of the HER family, inducing particularly important downstream oncogenic signaling. HER2 overexpression is used in breast cancer to select patients who may potentially respond to targeted therapy with an anti-HER2 agent; this concerns 15 to 30% of breast cancers. Targeted anti-HER2 therapies have significantly improved the prognosis of breast cancer patients [12].

The aim of this study was to assess HER2 protein overexpression and the secondary gene’s amplification using CISH (Chromogenic In Situ Hybridization)-improved NGS (Next-Generation Sequencing) p53-mutated endometrial carcinomas to determine if this could be a good approach for future studies on new targeted therapies to add to the therapeutical panel of endometrial cancer.

## 2. Materials and Methods

For this retrospective study approved by the Erasme University Hospital’s ethics committee (SRB2021316), the database of the Department of Pathology was studied. This database lists every gynecological case analyzed by the Department of Pathology. The cases concerning breast pathologies, ovarian pathologies, vulvar pathologies and cervix pathologies were not retained. Every endometrial case between 2019 and 2022 was analyzed to finally extract endometrial cancers meeting the following criteria:-The MMR and the p53 proteins analyzed by immunohistochemistry: MLH1 (MutL Homolog 1) (clone ES05, 1:50); MSH2 (MutS Homolog 2) (clone FE11, 1:50); MSH6 (MutS Homolog 6) (clone EP49, 1:100); and PMS2 (Postmeiotic Segregation 2) (clone EP51, 1:100) and p53 (clone DO-7, 1:200) [13].-Immunostaining was performed using a fully automated immunohistochemical system (Autostainer Link A48; Dako, Glostrup, Denmark).-The molecular analysis using Next-Generation Sequencing was then performed as we have previously described [14]. The hematoxylin-and-eosin-stained slides were read by the pathologists, who delimited the cellular tumor area and evaluated the tumor percentage, and was then used as a guide for a manual macrodissection. The percentage of tumoral cells in the samples ranged from 60 to 90%. The tumor tissue was manually macrodissected, scraped off the slide with a scalpel and transferred into a 1.5 mL tube, and then DNA was extracted from formalin-fixed paraffin-embedded tumor samples using the QIAamp FFPE Tissue Kit (Qiagen, Antwerp, Belgium). The DNA obtained was quantified using the Qubit^®^ Fluorometer in combination with the Qubit^®^ dsDNA HS Assay Kit (Life Technologies, Gent, Belgium). NGS was then performed in order to sequence hotspot mutations in the following panel of genes (Figure 1). Sequencing was performed on a PGM™ sequencer. The raw data were analyzed using the torrent suite software (v4.0 to v5.0—Life Technologies).

Once this selection was achieved, 118 cases between 2019 and 2022 were included in this series.

First, the proportions of histological types in the series were determined according to the WHO 2020 classification: low-grade endometrioid carcinomas, high-grade endometrioid carcinomas, serous carcinomas, clear cell carcinomas and carcinosarcomas; finally, other extremely rare histological types, such as neuroendocrine tumors, were excluded.

Next, the molecular profile protocoled via NGS was analyzed. The cases were classified according to the 4 tumoral subtypes identified by The Cancer Genome Atlas: pole-mutated; MMR-stable; MMR-instable; and p53-mutated.

The pole-mutated and MMR-instable tumors were excluded. Indeed, 8% of endometrial carcinomas with microsatellite instability and a third of Pole-mutated carcinomas are also carriers of a mutation in the TP53 gene [15].

However, it seems that these mutations are events encouraged by this initially present microsatellite instability and therefore do not affect the molecular profile or phenotype of these tumors because these mutations affect sequences of the TP53 gene that do not impact the function or expression of the p53 protein. These cases of unstable MMR tumors with a mutation of the added TP53 gene also have a better prognosis than tumors with a mutated P53 profile [16].

For the remaining cases with a TP53 mutation detected via NGS, the HER2 protein profile was studied.

The slides were again reviewed by two experienced pathologists in order to classify the cases according to their immunohistochemical classification, as seen in (Table 1) and Figure 2.

To finalize our description, in the cases with an immunohistochemistry “++” or “+++”, to determine if a gene was amplified, CISH (Chromogenic In Situ Hybridization) was performed [17]. The VentanaBenchMark Ultra Plus IHC/ISH (Roche Diagnostics) was used. Cell conditioning 2 was used for the pretreatment and CISH protease 3 for enzyme digestion and finally incubation with HER2 (dinitrophenol-labeled) and chromosome 17 (digoxigenin-labeled). The HER2 probes were incubated with antidinitrophenol antibody and horseradish-peroxidase-conjugated antibody and then silver reactions. The CEP17 probes were incubated with anti-digoxigenin antibody and alkaline-phosphatase-conjugated antibody, and then a red CISH Naphtol reaction was induced. Ventana hematoxylin II and bluing reagent were used to counterstain slides [18]. At least twenty nuclei, each containing red (Chr 17) and black (HER2), were enumerated (Figure 3). The HER2 status is based on the ratio formed by dividing the sum of HER2 signals for all nuclei divided by the sum of chromosome 17 signals. The amplification status is defined if the HER2/Chr17 ratio ≥2.0.

## 3. Results

### 3.1. Histological Type Proportions

A total of 68.64% of cases were low-grade endometrioid cancers; 10.17% were high-grade endometrioid cancers; 214.41% were serous cancers; 4.24% were clear cells cancers; and 2.54% were carcinosarcomas (Table 2).

### 3.2. Molecular Profile Proportions

Concerning the 118 cases included in the study (Figure 4), they were classified according to their molecular profile.

A total of 5.93% of cases had a Pole-mutated profile; 40.68% had an MMR-stable profile; 23.73% had an MMR-instable profile; and 29.66% had a p53-mutated profile (Table 3).

### 3.3. Proportions of p53-Mutated Cases According to the Histological Type

The 35 cases with a p53 protein mutation were classified according to their histological type. Altogether, 22.86% were low-grade endometrioid cancers; 11.43% were high-grade endometrioid cancers; 48.57% were serous cancers; 14.28% were clear cells cancers; 8.57% were carcinosarcomas (Table 4).

### 3.4. Proportions of the Different Histological Types Mutated for p53 Protein

Concerning the 81 low-grade endometrioid cancers in our series, 9.88% were p53-mutated; of the 12 high-grade endometrioid carcinomas, 33.33% were p53-mutated; of the 17 serous cancers, 88.24% were mutated; of the 3 clear cell cancers, 66% were mutated; and of the 3 carcinosarcomas, 100% were mutated (Table 5).

### 3.5. Proportions of Serous Endometrial Cancer Mutated for p53 Protein Mutation

A total of 88.24% of the 17 serous endometrial cancers of this series were mutated for the p53 protein; 5.88% were MMR-stable and 5.88% were MMR-instable (Table 6).

### 3.6. Proportions of HER2 Protein Immunohistochemistry Expression

Concerning the cases in this study that had a TP53 gene mutation detected using NGS, 1 of them was unavailable and the other 34 were analyzed by two experienced pathologists. Using the criteria previously described, 38.24% had a “0” profile for HER2 protein immunohistochemistry expression; 29.41% had a “+” profile; 23.53% had a “++” profile; and 8.82% had a “+++” profile (Table 7).

### 3.7. HER2 Gene Amplification by CISH

For the 11 cases in the series that had an immunohistochemistry profile corresponding to “++” or “+++” for the HER2 protein, 9 were analyzable. Four of them presented an amplification for the HER2 gene; four others presented an aneusomy, which looked like a CEP17 polysomy; and one case was not amplified.

With regard to the four cases presenting an amplification of the HER2 gene, 50% were clear cell carcinomas; 25% were carcinosarcomas and 25% were serous cancers (Table 8). The amplified HER2 cases were associated with different p53 gene mutations, including p.R342 exon 9; p.R248 exon 7; p.L265 exon 8; p.V216M exon 6, respectively.

Of the four other cases presenting an aneusomy for CEP17 (polysomy-like), 75% were serous cancers and 25% were carcinosarcomas (Table 9). The p53 gene mutations associated with these cases were p.H179Q exon 5; p.A129 exon 5; p.D281H exon 8; p.V218 del exon 6.

## 4. Discussion

We first aimed to answer: Is this series sufficiently representative of the distributions found in other series in order to be able to draw conclusions?

Regarding the proportions of histological types and molecular profiles, they followed the trends published in the literature; the endometrioid type affected 80% of patients while the others types of non-endometrioid carcinomas corresponded to 20% of tumors, with this group presenting mainly serous carcinomas [19]. A literature review reports a large majority of stable MMR tumors, which represented 39% of cases; a small percentage of mutated Pole tumors, around 7%; unstable MMR and mutated P53 profiles represented in similar proportions, in about 28% and 26% of cases, respectively [20]. Another literature review, conducted by Alexa et al., describes that Pole tumors are mutated in 4 to 12% of cases; 30 to 60% are stable MMR tumors; 23 to 36% of tumors have an unstable MMR profile; and finally, 8 to 24% of cases have a mutated P53 profile [21].

The second question is as follows: Is HER2 amplification worth studying in the future to add some new targeted therapies to the therapeutical panel of the p53-mutated endometrial cancer?

A randomized clinical trial that began in 2011 and is currently in phase II (NCT01367002) is investigating the therapeutic response of stage III or IV or recurrent serous carcinomas to carboplatin/paclitaxel compared to carboplatin/paclitaxel to which trastuzumab, an anti-HER2 agent, is added [22]. The first updated results in 2020 show that the addition of trastuzumab significantly improved progression-free survival and mean survival in women with advanced stage or recurrent HER2-positive cancer, with the best benefit observed in stage III or IV cancer.

Currently, it is recommended HER2 gene amplification be tested for only in serous histological types of endometrial cancer, which is considered the most aggressive histological type of endometrial cancer. However, Ross et al., in their retrospective study in 2021, report that in 94% of cases of endometrial cancers with amplification for the HER2 gene, there is a mutation of the TP53 gene as well [23]. This TP53 mutation was found in 88.24% of serous carcinomas in our study, but not in every carcinoma. Instead, in our series, it was found in 9.88% of low-grade endometrioid carcinomas, 33.33% of high-grade endometrioid carcinomas, 60% of clear cell carcinomas and 100% of carcinosarcomas. Because this TP53 mutation is not only found in serous carcinoma and because it is found in 94% of the cases presenting an HER2 gene amplification, research on HER2 gene amplification should not be exclusively reserved for serous carcinomas, as currently recommended.

Moreover, Ross et al. have observed at least one case of amplification of the HER2 gene in all histological types of endometrial cancer [23]. Instead, in our study, the serous type was not the only histological type for which we found an amplification of the HER2 gene. Indeed, this amplification was also present in carcinosarcomas and clear cell carcinomas and occurred in various mutational p53 profiles, including p.R342 exon 9; p.R248 exon 7; p.L265 exon 8; p.V216M exon 6 mutations.

Regarding the 21.57% of cases of mutated P53 tumors presenting a polysomal-like aneusomy for centromere 17, this is a phenomenon already described in breast cancers for which the clinical impact and effects on the expression of the HER2 protein are still uncertain [24].

A recent study showed that polysomy of centromere 17, without amplification of the HER2 gene, would be associated with a higher immunohistochemical expression of the HER2 protein, corresponding to “++”/”+++” scores [25]. Therefore, the importance of these tumors’ response to targeted anti-HER2 therapies arises.

Again, the data reported in the literature diverge. However, Hofmann et al., in 2008, had already described two cases of breast cancers overexpressing the HER2 protein in the presence of polysomy of centromere 17 and which presented a therapeutic response to trastuzumab [26]. Moreover, Sun et al. published similar results for their cohort with a therapeutic response observed for tumors with polysomy of centromere 17 when neoadjuvant chemotherapy containing an anti-HER2 agent was administered [25].

Of course, all these data concerning HER2 expression and/or amplification in endometrial cancers must be tempered, firstly, because endometrial tumors are often large (>30 mm) and HER2 expression/amplification may be heterogeneous. However, it has been shown that in other cancers (breast, stomach, etc.), this heterogeneity can affect not only the prognosis of these carcinomas but also their response to the various HER2 therapies. In addition, most of the clinical studies that tested the different anti-HER2 therapies are only based on the immunohistochemical score and do not consider HER2 amplification. Finally, in breast cancer, tumors considered HER2-negative (+ or ++ without HER2 gene amplification) may still respond to new HER2 therapies. Finally, in breast cancer, tumors that are considered HER2-negative may still respond to new HER2 therapies [27].

## 5. Conclusions

In this study, we have demonstrated that overexpression of HER 2 (++ or +++) was present in 31.4% of p53-mutated endometrial carcinomas and that among these, 36.3% showed HER2 gene amplification by CISH. This overexpression and gene amplification are not restricted to serous carcinomas but were also found in clear cell carcinomas and carcinosarcomas. These results suggest that various anti-HER2 agents could constitute interesting future treatments in this type of cancer. Naturally, our findings should be confirmed by larger studies in the future. Furthermore, in the era of personalized medicine, we advocate for the continuation of studies on targeted anti-HER2 therapies for cases with HER2 amplification. In addition, their extension to endometrial carcinomas with polysomal aneusomy for centromere 17 could be proposed, following studies already initiated for breast cancers.

## Figures and Tables

**Figure 1 cancers-15-01435-f001:**
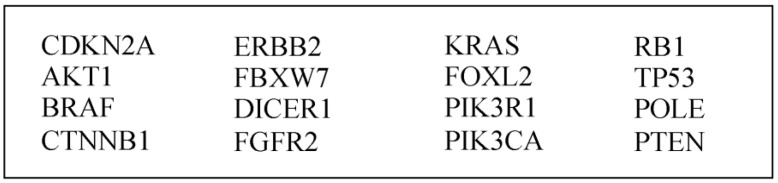
Genes tested via Next-Generation Sequencing (NGS).

**Figure 2 cancers-15-01435-f002:**
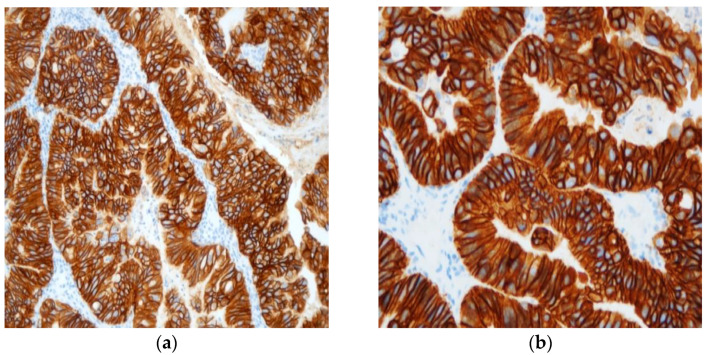
Typical example of +++ immunohistochemistry score for HER2 protein’s expression at ×20 (**a**) and ×40 (**b**) power fields.

**Figure 3 cancers-15-01435-f003:**
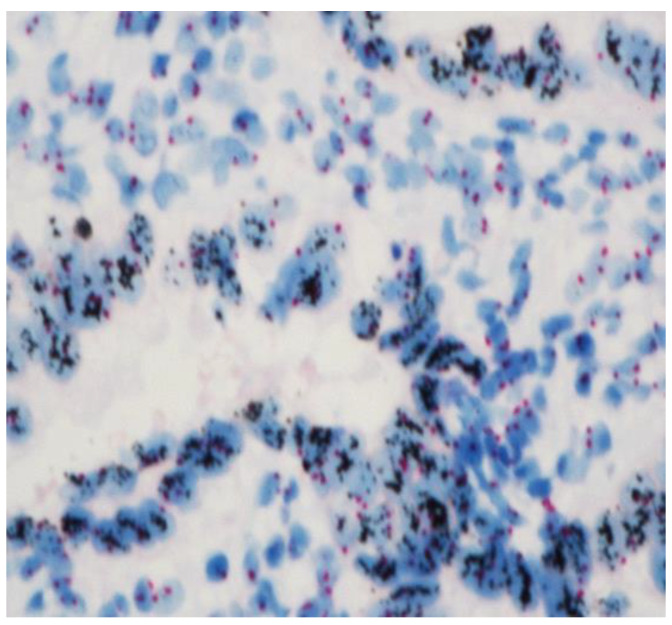
HER 2-amplified endometrial carcinomas with clusters of HER2 signals in black clearly contrasting with the 2 red signals (Chr 17) at ×40.

**Figure 4 cancers-15-01435-f004:**
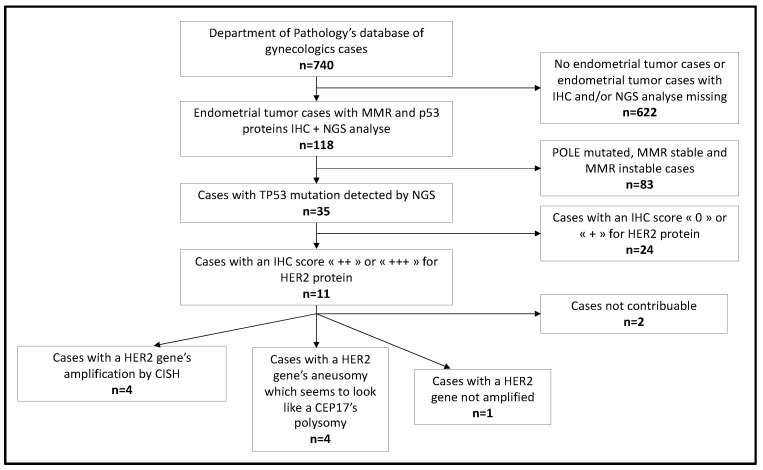
Flowchart of our study’s steps.

**Table 1 cancers-15-01435-t001:** HER2 protein immunohistochemistry classification criteria.

IHC Score	Criteria
0	No staining in tumor cells
+	Faint/barely perceptible, incomplete membrane staining in any proportion or weak in ≤10% of tumor cells
++	Intense or complete basolateral/lateral membrane staining in ≤30%, or weak to moderate in ≥10% of tumor cells
+++	Intense complete or basolateral/lateral membrane staining in ≥30% of tumor cells

**Table 2 cancers-15-01435-t002:** Histological type distribution.

Histological Type	Proportions
Low-grade endometroid	68.64% (*n* = 81)
High-grade endometrioid	10.17% (*n* = 12)
Serous	14.41% (*n* = 17)
Clear cell	4.24% (*n* = 5)
Carcinosarcomas	2.54% (*n* = 3)

**Table 3 cancers-15-01435-t003:** Molecular profile distribution.

Molecular Profile	Proportions
POLE-mutated	5.93% (*n* = 7)
MMR-stable	40.68% (*n* = 48)
MMR-instable	23.73% (*n* = 28)
P53-mutated	29.66% (*n* = 35)

**Table 4 cancers-15-01435-t004:** p53-mutated cases according to their histological type.

Histological Type	Proportions
Low-grade endometroid	22.86% (*n* = 8)
High-grade endometrioid	11.43% (*n* = 4)
Serous	48.57% (*n* = 17)
Clear cell	8.57% (*n* = 3)
Carcinosarcomas	8.57% (*n* = 3)

**Table 5 cancers-15-01435-t005:** Proportions of the different histological types mutated for p53 protein.

Histological Type	Proportions
Low-grade endometrioid	9.88%
High-grade endometrioid	33.33%
Serous	88.24%
Clear cell	66%
Carcinosarcomas	100%

**Table 6 cancers-15-01435-t006:** Serous endometrial cancer types according to molecular profiles.

Molecular Profile	Proportions
P53-mutated	88.24% (*n* = 15)
MMR-stable	5.88% (*n* = 1)
MMR-instable	5.88% (*n* = 1)

**Table 7 cancers-15-01435-t007:** Proportions of HER2 protein immunohistochemistry expression.

IHC Profile	Proportions
0	38.24% (*n* = 13)
+	29.41% (*n* = 10)
++	23.53% (*n* = 8)
+++	8.82% (*n* = 3)

**Table 8 cancers-15-01435-t008:** Amplified HER2 gene cases according to their histological type.

Histological Type	Proportions
Clear cell	50% (*n* = 2)
Carcinosarcomas	25% (*n* = 1)
Serous	25% (*n* = 1)

**Table 9 cancers-15-01435-t009:** Cases presenting an aneusomy for CEP17 (polysomy-like) according to their histological type.

Histological Type	Proportions
Carcinosarcomas	75% (*n* = 3)
Serous	25% (*n* = 1)

## Data Availability

Data are available at request from the authors.

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
