# Peer review of "HER2 Amplification in p53-Mutated Endometrial Carcinomas"

_cancers, 2023, doi:10.3390/cancers15051435_

Round 1

Reviewer 1 Report

This article by Belgian specialists in gynecology and breast pathology brings a new perspective on the treatment of endometrial cancers, suggesting new approaches to treatment. The authors make the point that rates of endometrial cancer are increasing in Belgium as they are in the rest of the world, emphasizing the need for expanded approaches to treatment. Their work relies on the Cancer Genome Atlas classification of four tumoral subtypes.

The authors bring significant and unique experience and expertise to the question. They base their suggestions on their analysis of 118 samples from the Erasme University Hospital 2019-2021 data set of 740 endometrial cases, 35 with TP53 mutation (~29% of the sample of 118), to suggest an expanded new avenue for the treatment. The p53 mutated profile was found in 88.25% the sample’s serous carcinomas.  Currently, testing for HER2 gene amplification is only done in serous histological types of endometrial cancer and response to anti-HER2 drugs is similarly confined.  Based on their findings, they suggest extending to histological types other than serous carcinomas, which have received emphasis in research and treatment because they are most aggressive. In this, the authors note that other types also respond to targeted therapies.

In light of increasing rates of endometrial and other gynecological cancers worldwide, transdisciplinary approaches to the problem are sorely needed, in part because they bring new perspectives and suggestions that they bring to the problem. This manuscript seems to do just that. Several suggestions include that:

·       The manuscript undergo extensive editing for English usage;

·       More acronyms be defined;

·       More detail be added on the data base at Erasme University Hospital. The description is very terse;

·       Discussion be added about the limitations of the sample and its implications for generalizability of findings beyond an exclusively Belgian sample.

I also would suggest that the conclusions, which are quite terse, be expanded. 

Author Response

All changes have been highlighted in blue in the revised version

- According to the remark 1 of the reviewer 1 :  

The English language has been improved.  

- According to the remark 2 of the reviewer 1 :

 All the acronyms have been clearly defined in the text.

- According to the remark 3 of the reviewer 1 :   

 More details have been added concerning the data base.

 Page 2,  lines 64  to 73 : “For this retrospective study… this panel of genes (Figure 1)”.

According to the remark 3 of the reviewer 1

We have included in the discussion/conclusion the limitation concerning the size of our samples.

 Page 9,  lines 243  to  246: “ Naturally our findings should be confirmed… for cases with HER 2 amplification”.

- According to the remark 5 of the reviewer 1 and  remark 6 of the reviewer 3:

The discussion and conclusions have been improved and expanded with more consideration of the different data clearly recovered in our work.

 Page 7, lines 205 to 215 : “Currently… recommended for now”.

Reviewer 2 Report

The authors in this article try to determine if the anti-HER2 targeted therapies could be used in the treatment of Endometrial cancer. 

I have a major question regarding the histological studies. The article suggests mutation in Tumor Protein 53 (TP53) might be the cause for serous carcinoma. I would urge the authors to highlight what and where is the mutation in the genomic sequence or protein sequence. For biochemistry or drug design modeling researchers, the answer to this question is important to develop targeted therapies to inhibit HER2 amplification.

Author Response

All changes have been highlighted in blue in the revised version.

- According to the comment 2 of the reviewer 2 :

The reviewer suggest the necessity to highlight what and where is the mutation in the genomic sequence or protein sequence. For biochemistry or drug design modeling researchers, the answer to this question is important to develop targeted therapies to inhibit HER2 amplification. We have mentioned the different p53 genomic mutations encountered in both HER 2 amplified and polysomic-like cases.:

Page 6: lines 181 to 184  :  “With regard…p.V216M exon 6” and page 6 : lines 189 to 192 : “ The four other cases… p.V218 del exon 6”.

- According to the comment 1 of the reviewer 2 :

We agree with this comment and clearly defined that p53 mutations are not restricted in serous carcinoma but also  in clear cell carcinomas, high grade endometrioid carcinomas  and carcinosarcomas.

Page 5 : lines 155 to 161: “Proportions of the … muted” and table 4.

Reviewer 3 Report

The researchers studied 118 samples of which Immunohistochemistry of P53 and MMR was available as well as NGS.

For 35 P53 mut tumor samples, Her2 expression was studied by Immunohistochemistry. 11 Samples rated as 2+ or 3+ staining were then analysed bij CISH and found positive fore gene amplification in 4 and aneusomy (CEP17 polysomy -like) in 4.

Of these 35 P53 mut tumors 15 were serous tumours.

Of serous tumours, 88% were P53 mut.

In the simple summary I read: “The aim of this study is to determine if the anti-HER2 targeted therapies could be used in the treatment of this cancer, following the therapeutic schedule already used in the breast cancer.” This aim is not at all accomplished with these results.

In the abstract I read: “In conclusion, the response to anti- HER2 drugs should be studied for histological types other than serous carcinomas. This conclusion is not supported by the results:  In the result section table 3 shows the proportion of different histological types endometrial cancer with P53 mutations. It is not stated how many non-serous P53 mutated cases ultimately have HER2 amplification.

The abstract also states: “Further investigations on the different types of anti-HER2 agents would need to be carried out. Finally, the future studies could be extended to endometrial carcinomas with polysomal aneusomy for centromere 17, following studies initiated for breast cancers”This again is a conclusion that bears little relationship with the results.

In the discussion is stated ( 158):

“In our study, the mutation in the TP53 gene is not exclusively found in serous types of endometrial cancer. It is found in 9.88 % of low- 159 grade endometrioid carcinomas, 33.33% of high-grade endometrioid carcinomas and 160 100% of clear cell carcinomas and carcinosarcomas”

These are results that are not provided in the results section nor is described in the methods section how these percentages are assessed.

Large Parts of the discussion are about the treatment of her2 over expressing tumours which is not the subject of the research. For example (164):

“With regard to the responses of the tumors to the Trastuzumab as anti-HER2 agent, several cases of resistance to trastuzumab in endometrial cancers have been reported [14,15]. Ross et al. report that certain genetic alterations commonly found in amplified HER2 cases, such as PIK3CA activation, are associated with decreased response to Trastuzumab. One of the phenomena highlighted to explain this resistance is the for- mation of HER2/HER3 heterodimers [16]. Therefore, agents inhibiting this heterodimeri- zation such as Pertuzumab seem to be a pathway to be studied in the future.  Regarding the 21.57% of cases of mutated P53 tumors presenting a pol….”

In several places in the text P53 muted is used instead of P53 mutated

Author Response

All changes have been highlighted in blue in the revised version.

- According the general comment of the reviewer 3 concerning this paper:

“refocusing on the data we found”. We concur with this point of view and we clearly and simply refocused on the data found in this work: the simple summary/abstract/conclusions have been clearly modified in this sense.

Page 1: lines 11 to 27: “the endometrial cancers… in these variants of aggressive carcinomas”.

 Page 7 : lines 238 to 248 : “In this study… for breast cancers”.

- According to the comment 3 of the reviewer 3 :

“It is not stated how many non-serous P53 mutated cases ultimately have HER2 amplification”. The proportion HER2 amplified and polysomic-like cases according to their histological types was clarified.

Page 6 lines 182 to 195 and tables 7 and 8 : “With regard to… exon 6”.

- We put in considerations the comment 4 of the reviewer 3 and we have drastically reduced the therapeutic considerations in the discussion.

- According to the comment 5 of the reviewer 3 :

The results have been clarified in tables 4, 7, 8 (pages 5 and 6).

- According to the comment 5 of the reviewer 3 :

The methodology and in particular CISH technique and HER 2 interpretation scoring and HER 2 illustration have been clarified

 Page 3: lines 106 to 177: “In the cases…by dividing”, page 4: line 118 to 124 : “the sum … Chr 17”.

- According to the comment 6 of the reviewer 3 :

Muted has been used instead of mutated in all the text.

- According to the remark 5 of the reviewer 1 and  remark 6 of the reviewer 3:

The discussion and conclusions have been improved and expanded with more consideration of the different data clearly recovered in our work.

 Page 7, lines 205 to 215 : “Currently… recommended for now”.

Round 2

Reviewer 2 Report

Thank you for addressing my comments and for improving your manuscript. I recommend in favour of the article to be accepted for publication.